# Clinical Management and Therapy of Precocious Puberty in the Sapienza University Pediatrics Hospital of Rome, Italy

**DOI:** 10.3390/children10101672

**Published:** 2023-10-10

**Authors:** Ginevra Micangeli, Roberto Paparella, Francesca Tarani, Michela Menghi, Giampiero Ferraguti, Francesco Carlomagno, Matteo Spaziani, Ida Pucarelli, Antonio Greco, Marco Fiore, Luigi Tarani

**Affiliations:** 1Department of Maternal Infantile and Urological Sciences, Sapienza University of Rome, Viale del Policlinico 155, 00161 Rome, Italyroberto.paparella@uniroma1.it (R.P.);; 2Department of Experimental Medicine, Sapienza University of Rome, Viale del Policlinico 155, 00161 Rome, Italyfrancesco.carlomagno@uniroma1.it (F.C.); matteo.spaziani@uniroma1.it (M.S.); 3Department of Sensory Organs, Sapienza University of Rome, 00185 Rome, Italy; 4Institute of Biochemistry and Cell Biology, IBBC-CNR, 00185 Rome, Italy

**Keywords:** precocious puberty, pseudopuberty, children, pediatrician, GnRh analogs

## Abstract

Puberty identifies the transition from childhood to adulthood. Precocious puberty is the onset of signs of pubertal development before age eight in girls and before age nine in boys, it has an incidence of 1/5000–1/10,000 with an F:M ratio ranging from 3:1 to 20:1. Precocious puberty can be divided into central, also known as gonadotropin-dependent precocious puberty or true precocious puberty, and peripheral, also recognized as gonadotropin-independent precocious puberty or precocious pseudopuberty. Thus, the main aim of this narrative report is to describe the standard clinical management and therapy of precocious puberty according to the experience and expertise of pediatricians and pediatric endocrinologists at Policlinico Umberto I, Sapienza University of Rome, Italy. In the suspicion of early sexual maturation, it is important to collect information regarding the age of onset, the speed of maturation of secondary sexual features, exposure to exogenous sex steroids and the presence of neurological symptoms. The objective examination, in addition to the evaluation of secondary sexual characteristics, must also include the evaluation of auxological parameters. Initial laboratory investigations should include serum gonadotropin levels (LH and FSH) and serum levels of the sex steroids. Brain MRI should be performed as indicated by the 2009 Consensus Statement in all boys regardless of chronological age and in all girls with onset of pubertal signs before 6 years of age. The gold standard in the treatment of central precocious puberty is represented by GnRH analogs, whereas, as far as peripheral forms are concerned, the triggering cause must be identified and treated. At the moment there are no reliable data establishing the criteria for discontinuation of GnRH analog therapy. However, numerous pieces of evidence suggest that the therapy should be suspended at the physiological age at which puberty occurs.

## 1. Introduction

Puberty is the transitional period from childhood to adulthood characterized by major physical and psychological modifications leading to the development of secondary sexual characteristics, the maturation of the gonads and the achievement of reproductive capacity [1]. Puberty is a complex process characterized by environmental, genetic, geographical and metabolic factors [2]. The mechanism underlying pubertal activation remains unknown, although the following have been identified as possible causes: adrenal activation, physical and psychological stress, an abundance of adipose tissue and the inflammation of the intestinal tract [3,4].

Puberty begins with the activation of the hypothalamic-pituitary-gonadal axis (HPG), which already occurs during fetal life, but is usually silenced in the final period of pregnancy and then reactivated immediately after birth [5,6]. This post-natal transitory activation is defined as mini-puberty and lasts up to 6 months in boys and up to 2 years in girls, until the blockage of gonadotropin-releasing hormone (GnRH) secretion, which will resume during puberty [7,8].

The beginning of puberty is determined by the secretion of GnRH at the hypothalamic level, which in turn activates the production of two hormones by the pituitary gland, the gonadotropins luteinizing hormone (LH) and follicle-stimulating hormone (FSH) [9]. LH and FSH act on the gonads, promoting gametogenesis [10]. Pubertal development is considered physiological when it begins between the ages of 8 and 13 in females and between 9 and 14 years in males, although it is a variable process within each individual, lasting on average between 3 and 5 years [2,10].

### 1.1. Precocious Puberty

Precocious puberty (PP) is the onset of secondary sexual features before the age of eight in girls and before the age of nine in boys. Specifically, the first sign of pubertal activation is represented by thelarche in females and an increase in testicular volume in males greater than or equal to 4 mL [1,10]. It is a relatively rare condition affecting 1:5000–1:10,000 children, with an F:M ratio ranging from 3:1 to 20:1 [11,12]. There has been an increase in diagnoses of precocious puberty partly attributable to the SARS-CoV-2 pandemic, probably related to a sedentary lifestyle characterized by being overweight, the use of electronic devices and stress-related symptoms acting as endocrine disruptors [13,14,15,16,17].

PP can be classified as follows:-central or true precocious puberty (CPP), if it is determined by early activation of the HPG axis with the production of gonadotropins;-peripheral or precocious pseudopuberty (PPP), unrelated to the production of gonadotropins.

### 1.2. Central Precocious Puberty

CPP accounts for about 80% of all forms of PP and is caused by early activation of the HPG axis with increased GnRH secretion and gonadal activation [3,18]. Although rarer, CPP in males is more often related to underlying hypothalamic-pituitary organic lesions [7]. Although it is often idiopathic, numerous genetic mutations related to CPP have been identified, among these, the loss of function mutation of the *MKRN3* gene is one of the most involved [12]. The loss of function of the *MKRN3* gene, located within the Prader-Willi syndrome region on chromosome 15q11.2, is responsible for a stimulatory action on GnRH secretion [19,20].

CPP could have a familial form in almost one quarter of the children [21]. The detection of this inherited condition increased after the discovery of autosomal dominant CPP with paternal transmission due to mutations also in the DLK1 gene [21]. Indeed, it has been shown that the incidence of familial CPP was disclosed at 22%, with a comparable frequency of paternal and maternal transmission [21]. Lineage analyses of families with maternal transmission indicated an autosomal dominant inheritance. Clinical and hormonal characteristics, as well as the effects of GnRHa administration, were similar among individuals with dissimilar forms of transmission of familial CPP [21]. MKRN3 loss-of-function mutations were the most predominant source of familial CPP [21], followed by DLK1 loss-of-function mutations, affecting, respectively, 22% and 4% of the analyzed families; both forms affected exclusively families with paternal transmission [21]. Rare variants of indeterminate significance were discovered in CPP families with maternal transmission [21].

Other genetic mutations have been associated with CPP, including gain-of-function mutations in the *KISS1* gene, which encodes the kisspeptin proteins capable of stimulating the production of gonadotropins, and the *KISS1R* gene, formerly known as *GPR54,* which encodes its G protein-coupled receptor [22]. Among the most important factors in the development of precocious puberty, we certainly find endocrine disruptors that significantly increase the risk of precocious puberty [3]. One of the most studied factors would be leptin, which would seem to increase with the increase in visceral fat causing an early release of GnRH [15].

Another peptide involved would appear to be ghrelin, a peptide with orexigenic action produced in the stomach, which instead has an inhibitory action on the production of GnRH by decreasing the responsiveness of LH to its release factor [10] and simultaneously increasing pituitary growth hormone (GH) secretion, thus acting at the intersection of gonadotropic and somatotropic axes [23].

In recent years, an association has emerged between pubertal timing and exposure to environmental factors that would cause an early activation of the pubertal axis [24,25]. Some of these substances act as estrogen receptor agonists or androgen receptor antagonists [7,22,26,27]. These endocrine-disrupting chemicals have been suggested as affecting the age of pubertal onset and include phthalates, pesticides, isoflavonoids, phytoestrogens, polybromobiphenyl and mycotoxins [11,28].

### 1.3. Peripherical Precocious Puberty

PPP is characterized by an increase in adrenal and gonadal sex steroids in the absence of HPG axis activation; the pubertal characteristics may be valid for the child’s sex (isosexual) or inappropriate, with virilization of girls and feminization of boys (contrasexual) [29]. It can be congenital, with the most frequent forms represented by congenital adrenal hyperplasia and McCune–Albright syndrome, or acquired, mainly related to hormone-secreting endocrine tumors [2,11,30].

The main goal of this work is to display the standard clinical management and therapy of precocious puberty according to the experience and expertise of the pediatric endocrinology of Policlinico Umberto I, the Sapienza University of Rome, Italy where approximately 50 children and adolescents are treated each year, equally distributed between the sexes.

## 2. Diagnosis

Regardless of the cause, PP must be promptly recognized since it is associated with accelerated stature growth and skeletal maturation if untreated, inducing an early growth arrest with premature epiphyseal fusion due to excess sex steroids, which sometimes results in short adult height [2]. The key features suggesting PP are the progressive development of breasts in girls and testicular volume in boys over a short period of observation (3–6 months) associated with rapid height growth (height velocity >6–7 cm/year) especially in girls aged between 6 and 8 years, when this condition occurs more frequently [5,31,32].

The family pediatrician plays a fundamental role in paying attention to pubertal development during normal health checks of the child, quickly referring a patient to a pediatric endocrinologist in case of pubertal activation signs, considering its increased prevalence in European countries [30,33].

In the suspicion of early sexual maturation, in kids presenting signs of secondary sexual development before the age of eight (females) or nin (males), the evaluation should begin with an accurate medical and familial history as shown in Figure 1 [22,34]. It is important to collect information regarding age of onset, rate of maturation of secondary sexual characteristics, exposure to exogenous sex steroids and the presence of neurological symptoms [10,14].

The physical examination, in addition to the evaluation of secondary sexual characteristics, must also include the evaluation of auxological parameters, such as weight, height, body mass index and height velocity (cm/year) [12,34]. Growth spurt is an important feature of pubertal development. In fact, growth acceleration with growth centile change supports the diagnosis of pubertal activation and therefore of PP [5,26].

Evaluation of bone age should be performed because children with PP frequently show advanced bone age, greater than two standard deviations beyond chronological age [35,36,37]. However, an advanced bone age does not rule out a benign pubertal variant, since up to 30% of kids with benign premature adrenarche have bone ages ≥2 years in advance of their chronologic age [25,38,39].

The physical analysis should include the assessment of visual fields (given the possibility of a central nervous system (CNS) lesion) and the examination for café-au-lait spots (suggestive of McCune–Albright syndrome or neurofibromatosis) [29,40].

Marshall and Tanner’s criteria are used to assess the stage of pubertal development as illustrated in Table 1 [41]. Furthermore, it may be difficult to distinguish precocious puberty from benign variants of precocious pubertal development such as isolated premature thelarche, premature adrenarche and lipomastia [1].

In girls under two years of age, the finding of isolated thelarche is frequent and is rarely associated with a pathological condition. Isolated premature thelarche is a benign, self-limiting condition characterized by the appearance of breast tissue in the absence of other signs indicative of pubertal development [12]. Transient thelarche also appears to be a frequent and benign phenomenon, characterized by peculiar subsequent pubertal timing and pathway, apparently unrelated to HPG axis activation or adiposity [42].

Premature adrenarche is also a benign condition, characterized by the appearance of acne, axillary hair and pubic hair that occurs in girls under 8 years of age and in boys under 9 years of age, in the absence of other signs of pubertal development and is caused by adrenal androgen secretion in the absence of activation of the HPG axis [22]. Furthermore, in obese or overweight girls it is important to distinguish a condition of lipomastia, an accumulation of adipose tissue in the breast area, from the thelarche, which is identified with the presence of glandular breast tissue below the areola [43].

Initial laboratory investigations should include serum gonadotropin (LH and FSH) levels and sex steroids, estradiol in girls and testosterone in boys [44,45,46]. Baseline LH > 0.3 mIU/mL is considered diagnostic for central precocious puberty; however, values below this limit do not exclude the diagnosis and require further diagnostic investigations. Measurable estradiol values or testosterone values >30 ng/dL suggest but do not confirm the diagnosis [47,48]. Therefore, children with clinical signs of early pubertal development and baseline LH values <0.3 mIU/mL are candidates for GnRH stimulation testing to identify HPG axis activation. Peak LH values >5 mIU/mL are considered indicative of pubertal activation. FSH values alone are not diagnostic, but the post-stimulus LH/FSH ratio is useful as an indicator of increased LH secretion using a cut-off between 0.6 and 1 [6,19]. However, the sensitivity and specificity of this index are lower than using the isolated post-stimulus LH surge.

In contrast, suppressed FSH and LH values associated with increased sex steroids suggest the diagnosis of PPP. In these cases, it is essential to complete the diagnostic procedure with the measurement of tumor markers (alpha 1-fetoprotein, beta-HCG, CEA and CA125), in order to exclude hypersecretion of sex steroids of a neoplastic or paraneoplastic nature, and the measurement of serum dehydroepiandrosterone sulfate (DHEA-S) and 17-hydroxyprogesterone levels is advisable, which may be increased in adrenal tumors or congenital adrenal hyperplasia due to 21-hydroxylase deficiency [11,49].

In the diagnostic process of suspected precocious puberty, it is important to exclude an unknown and therefore untreated hypothyroidism condition, mainly if there is slow instead of rapid growth and clear hypothyroid signs and symptoms [50]. Initial evaluation of the child with suspected precocious puberty should include the assessment of bone age, because children with precocious puberty frequently have advanced bone age, greater than two SDs of chronological age [35,36,37].

Furthermore, pelvic ultrasound, a quick, non-invasive and low-cost examination, is a useful support for diagnosis by evaluating uterine development and ovarian volume and investigating the presence of ovarian cysts or tumors [7,51]. During infancy, the ovarian volume is stable, the fundus of the uterus and the cervix have a similar width and assume a tubular configuration, while in the pubertal phase, the uterus increases in volume and the cervix assumes the typical pear shape of adulthood [11,52].

The following ultrasound criteria aid in correctly identifying PP:-Uterine longitudinal diameter ≥3.4 cm [53].-Uterine volume ≥1.8 mL (specific for pubertal onset) [54].-Ovarian volume ≥3 mL (or 3.4 mL for girls between 6 and 8 years) [55].

The presence of an endometrial stripe on pelvic ultrasound is also indicative of precocious puberty [56,57]. Overall, the ovarian volume represents the best indicator of PP, whereas uterine length is more capable of differentiating isolated premature thelarche from premature puberty [55]. A uterine fundal/cervical ratio ≥1 is generally inaccurate and no longer used [55].

Ultrasound examination also aids in the evaluation of boys with suspected PP, in which testicular enlargement represents the first sign of HPG axis activation. Specifically, testicular ultrasound represents the gold standard technique for the assessment of testicular volume [58], with a cut-off of 2.7 mL (calculated employing Lambert’s formula) corresponding to the traditional criteria of 4 mL defining Tanner stage II [59]. Furthermore, testicular ultrasound can predict testicular function [60] and is crucial for the evaluation of cryptorchidism and causes of scrotal enlargements, such as inguinoscrotal hernias, hydrocele, epididymal-orchitis, varicocele, testicular cancer and oncohematological disorders [58].

In patients for whom the diagnosis of PPC is reached, it is important to exclude the presence of organic pathology affecting the CNS [61]. Among the main risk factors suggesting the presence of organic brain lesions are male sex, early age of onset, rapid progression and the presence of neurological signs and symptoms. Brain MRI should also be considered in all children with rapidly progressive CPP [62]. However, numerous studies have shown that the detection of CNS tumors in girls aged between 6 and 8 years with PPC is a rare occurrence. Brain MRI in all boys regardless of chronological age and in all girls with onset of pubertal signs before 6 years of age should be considered. To date, it is still under discussion whether to perform this instrumental investigation in girls aged between 6 and 8 years, especially in those with a normal sequence of pubertal development and no clinical evidence of CNS lesions, in particular in the presence of a family history of earlier pubertal onset [4].

Genetic analysis should be considered in the presence of a family history of CPP or in the event that clinical features attributable to syndromic forms are present. To date, the mutation in the MKRN3 gene represents the most frequent form of monogenic CPP, with a prevalence of 33–47% in forms with familial recurrence and 0.4–5% in sporadic cases [1,35]. Furthermore, CPP is a key feature of several genetic syndromes such as Silver–Russell syndrome, Williams–Beuren syndrome, and Temple syndrome [12,19,63,64]

A pelvic ultrasound in females with PPP should be performed to identify the presence of an ovarian cyst or tumor, while in males an ultrasound examination of the testes to rule out a Leydig cell tumor is essential. In both females and males, peripheral precocity and progressive virilization and/or markedly elevated serum adrenal androgens may be due to an adrenal tumor, thus a computed tomography or MRI of the adrenal glands should be performed whenever other conditions such as congenital adrenal hyperplasia and exogenous androgen exposure have been excluded [65].

## 3. Treatment

Treatment of PP aims at preserving growth potential, synchronizing pubertal development with peers and improving psychological distress [11,12]. The main clinical criterion for initiation of therapy is the finding of pubertal progression, in children under the age of eight (females) or nine (males), with growth acceleration confirmed in a 3–6 month follow-up period [66,67,68]. This observation period may not be necessary if the bone age is markedly advanced or if the girl or boy presents with Tanner stage III [41,43]. The treatment is also indicated if PP is responsible for psychological and psychosocial disorders that can compromise the quality of life of patients and cause emotional and behavioral disorders that can also be detected at later ages [1,69].

The gold standard in the CPP treatment is represented by GnRH analogs (GnRHa) [70]. Their rationale for use is based on the recognition that, after an initial transient stimulation of gonadotropin secretion from the pituitary (termed “flare up”), high concentrations of GnRH eventually cause a complete, but reversible, suppression of the HPG axis by down-regulating the GnRH receptor, consequently inhibiting the secretion of gonadotropins [10,12]. GnRHa is available in different formulations: although slow-release formulations administered monthly were previously the most frequently used, formulations administered every 3 or 6 months (leuprolide and triptorelin) have been introduced in recent years, as well as subcutaneous implants of histrelin capable of inducing suppression of the hypothalamic-pituitary-gonadal axis for a period of 12–24 months [1,66,71,72]. As regards the therapeutic dosage required for the suppression of the HPG axis, there is no univocal opinion; in the USA, higher dosages are used (7.5 mg/month) while, in Europe, the monthly dosage used is 3.75 mg every 28 days [72,73]. Table 2 shows the different available formulations of GnRHa.

Currently, there are no firm data establishing criteria for discontinuation of GnRHa therapy [7,75,76]. However, numerous pieces of evidences suggest that the therapy should be suspended at the mean physiological age in which puberty occurs (between 10.5 and 11.5 years in females and between 12 and 13 years in males), or when a bone age of around 12 years in girls and 13 years in boys is reached or in cases of a marked reduction of growth velocity during therapy [11,75]. Thus, the decision to discontinue therapy is individualized and is based on many specific patient features, including predicted and absolute height, chronological age, pubertal stage, psychosocial factors and family preferences [3,77].

Major factors affecting height prognosis include timely initiation of treatment, age at onset of puberty, bone age and height at diagnosis and target height [33,43]. Girls who start treatment before the age of 6 have better outcomes than patients who start treatment between 6 and 8 years while starting therapy after 8 years of age does not appear to be associated with an increase in height in adulthood [1,11,71]. GnRHa therapy is generally well tolerated in childhood, although the most frequently described adverse events are headache, injection site reactions and hot flushes, which in most cases occur early and are resolved by subsequent GnRHa administrations [11,78,79,80]. More rare is the occurrence of vaginal discharge, or the development of a sterile abscess at the injection site or at the subcutaneous implant site, which may lead to loss of efficacy of the therapy [3,76]. Weight gain has been documented during therapy in some patients; however, the data available to date indicate that long-term GnRHa treatment does not affect body composition and the onset of obesity or the increased incidence of polycystic ovary syndrome (PCOS) in adulthood [81,82].

When CPP is caused by a CNS lesion, therapy is also directed toward the underlying pathology when possible. Treatment of PPP is instead aimed at blocking the secretion and/or response to the sex steroids, since PPP does not respond to GnRHa therapy, and therefore varies according to the underlying condition. In case of exposure to exogenous sex steroids, the source must be identified and removed.

Tumors of the testis, adrenal gland or ovary are often treated with the integration of surgery, radiotherapy and chemotherapy when necessary, depending upon the site and histologic type. A functioning ovarian follicular cyst, which is the most common cause of peripheral precocity in females, usually appears and regresses spontaneously, so conservative management with a clinical and ultrasonographic follow-up is usually chosen. Larger cysts may require surgery because they can predispose to ovarian torsion [83].

Glucocorticoids are needed in classic congenital adrenal hyperplasia. Anyway, the treatment of this and other defects in adrenal steroidogenesis is beyond the scope of this article. The McCune–Albright syndrome is defined as the triad of fibrous dysplasia of bone, café-au-lait skin pigmentation and peripheral sexual precocity. It is caused by a somatic activating mutation in *GNAS* (a gene encoding the alpha subunit of the Gs heterotrimeric G protein) which leads to continued stimulation of endocrine function [84,85]. PPP is much more common in females, because of the development of recurrent ovarian cysts, resulting in estradiol production and intermittent vaginal bleeding. Affected males produce elevated testosterone, but PPP is less likely. Treatment with aromatase inhibitors is commonly used for PPP in this syndrome [86].

Children might develop secondary CPP if exposed to high serum sex steroid levels, once the cause of PPP has been treated, therefore requiring continuous clinical monitoring for signs of progressive breast or testicular development and, if necessary, GnRHa addition [87].

## 4. Discussion and Conclusions

Puberty is a multifaceted process of transition from childhood to adulthood and its mechanisms are still not well known. PP is more common in females and more frequently it concerns idiopathic forms of CPP, but the recent characterization of genes involved in pubertal development underlines the important role of these factors in determining pubertal timing. It is important to identify the child with pathological pubertal development in order to undertake an accurate diagnostic and therapeutic procedure. The main goals of the treatment of precocious puberty are the preservation of growth potential, the synchronization of pubertal development with peers and the improvement of psychological distress. PPP is an extremely heterogeneous condition and can represent a manifestation of numerous pathologies, therefore an accurate etiological diagnosis is essential for correct management.

## Figures and Tables

**Figure 1 children-10-01672-f001:**
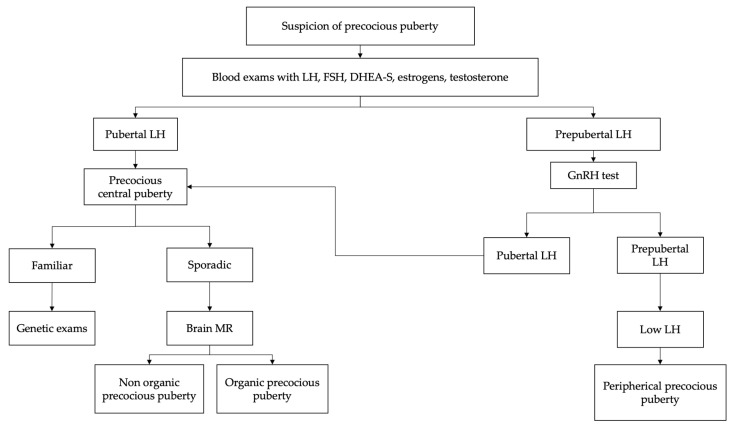
Suggested flowchart for the management of precocious puberty [8,11].

**Table 1 children-10-01672-t001:** The table shows the Tanner stages of puberty for females and males [11,41].

Tanner Stage	Breast (Female)	Pubic Hair (Female, Male)	Genitalia (Male)
I	Preadolescent	None	Preadolescent
II	Breast bud palpable under the areola	Sparse, long, straight	Enlargement of scrotum/testes
III	Breast tissue palpable outside areola; no areolar development	Darker, curling, increased amount	Penis grows in length; testes continue to enlarge
IV	Areola elevated above the contour of the breast, forming a “double scoop” appearance	Coarse, curly, adult type	Penis grows in length/breadth; scrotum darkens, testes continue to enlarge
V	The areolar mound regresses into a single breast contour with areolar hyperpigmentation, papillae development, and nipple protrusion	Adult, extend to thighs	Adult shape/size

**Table 2 children-10-01672-t002:** The table shows the different types and formulations of GnRHa [1,10,74].

GnRHa:	Duration of the Depot:	Dose:
Leuprolide	1 month	3.75 mg
		7.5 mg
		11.25 mg
		15 mg
	3 months	11.25 mg
		30 mg
Triptorelin	1 month	3.75 mg
		11.25 mg
	6 months	22.5 mg
Histrelin	12–24 months	50 mg (60 µg/die)

## Data Availability

Not applicable.

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
