# Peer review of "Clinical Management and Therapy of Precocious Puberty in the Sapienza University Pediatrics Hospital of Rome, Italy"

_children, 2023, doi:10.3390/children10101672_

Round 1
Reviewer 1 Report
Dear Authors,
I would like to express my appreciation for your submission entitled "Clinical Management and Therapy of Precocious Puberty in the 2 Sapienza University Pediatrics Hospital of Rome, Italy".
Your research addresses an important and timely issue, and the paper is well-written, clear, and organized. The methodology is sound, and the results provide valuable insights into the subject matter.
I believe that the paper, as it stands, will make a valuable contribution to our field and will be of interest to our readership. Therefore, I recommend that it be accepted for publication without the need for further revisions.
Author Response
Dear Authors,
I would like to express my appreciation for your submission entitled "Clinical Management and Therapy of Precocious Puberty in the 2 Sapienza University Pediatrics Hospital of Rome, Italy".
Your research addresses an important and timely issue, and the paper is well-written, clear, and organized. The methodology is sound, and the results provide valuable insights into the subject matter.
I believe that the paper, as it stands, will make a valuable contribution to our field and will be of interest to our readership. Therefore, I recommend that it be accepted for publication without the need for further revisions.
Reply: We do thank the reviewer for the positive comments
Reviewer 2 Report
I think this report is a very concise review. So, l do not have any request.
Author Response
I think this report is a very concise review. So, l do not have any requests.
Reply: many thanks
Reviewer 3 Report
This is a comprehensive review of CPP and its variants which will be useful to readers who are not already familiar with the subject matter, and explains clearly how one center in Italy evaluates patients with suspected PP.
Specific comments:
1. The abstract is too long and the first 7 lines can be condensed
2. The introduction is also too long and the first 3 paragraphs can be shortened as they do not deal with management issues which should be the main focus.
3. Iine 135: "which SOMETIMES results in short adult height". Many patients with CPP are tall at the time of diagnosis and early cessation of growth will bring their height into the normal range.
4. Line 136: height velocity >6-7 cm/year
5. Figure 1: top box should read "suspicion of PP". Also in several places they need to replace puberal with pubertal.
6. Line 194: in contrast, suppressed LH and FSH (not just FSH)
7. Line 196: with the measurement (not dosage) of tumor markers
8. Line 202: it is important to exclude an unknown and therefore untreated hypothyroidism condition, mainly if there is slow instead of rapid growth and clear hypothyroid signs and symptoms. (delete "especially if there is a poor activation of the pituitary gonadotropins after a stimulus test ')
Noted above
Author Response
Reviewer 3
This is a comprehensive review of CPP and its variants which will be useful to readers who are not already familiar with the subject matter, and explains clearly how one center in Italy evaluates patients with suspected PP.
Reply: We sincerely appreciated the constructive comments of the reviewer. As you will see, according to your comments, we modified the text of the revised paper highlighting in light yellow the modifications we made.
Specific comments:
1 The abstract is too long and the first 7 lines can be condensed
Reply: as requested, the abstract was shortened.
2 The introduction is also too long and the first 3 paragraphs can be shortened as they do not deal with management issues which should be the main focus.
Reply: According to your comment and also to the requests of the other reviewers and editors, we modified the Intro.
3 Line 135: "which SOMETIMES results in short adult height". Many patients with CPP are tall at the time of diagnosis and early cessation of growth will bring their height into the normal range.
Reply: We do apologize for the typing mistake. We made corrections (now line 131).
4 Line 136: height velocity >6-7 cm/year
Reply: As suggested, we modified the text (now line 134)
5 Figure 1: top box should read "suspicion of PP". Also in several places they need to replace puberal with pubertal.
Reply: We are quite grateful for this comment aimed at improving the quality of the pict.
6 Line 194: in contrast, suppressed LH and FSH (not just FSH)
Reply: As suggested, we modified the text (now line 193)
7 Line 196: with the measurement (not dosage) of tumor markers
Reply: As suggested, we modified the text (now line 195)
8 Line 202: it is important to exclude an unknown and therefore untreated hypothyroidism condition, mainly if there is slow instead of rapid growth and clear hypothyroid signs and symptoms. (delete "especially if there is a poor activation of the pituitary gonadotropins after a stimulus test ')
Reply: As suggested, we deleted the sentence (now lines 201-203)
Reviewer 4 Report
It is a useful summary of precocious puberty diagnostics and management, especially for the junior clinicians. I would recommend to review more the genetic possibilities of PPP, such as genetic background of adrenocortical tumours and the genetic syndromes. Furthermore, there is lack of management and follow-up for PPP presented. Please use italic for genes name. Presented gonadotrophins values are probably measured by highly sensitive methods, it should be discussed further.
Moderate editing of English language required
Author Response
Reviewer 4
It is a useful summary of precocious puberty diagnostics and management, especially for the junior clinicians. I would recommend to review more the genetic possibilities of PPP, such as genetic background of adrenocortical tumours and the genetic syndromes. Furthermore, there is lack of management and follow-up for PPP presented. Please use italic for genes name. Presented gonadotrophins values are probably measured by highly sensitive methods, it should be discussed further.
Reply: We do thank the reviewer for the productive comments. Indeed, according to your suggestions and also to the comments of the other reviewers we made changes throughout the revised text highlighting them in light yellow.